# Empowering Student Pharmacists to Counsel Patients on Endocrine Disrupting Chemicals through Interactive Role-Play

**DOI:** 10.3390/pharmacy12020055

**Published:** 2024-03-26

**Authors:** Alina Cernasev, Amy Hall, Stacey Thomas-Gooch, Devin Scott

**Affiliations:** 1Department of Clinical Pharmacy and Translational Science, College of Pharmacy, University of Tennessee Health Science Center, Nashville, TN 37211, USA; sthoma36@uthsc.edu; 2Teaching and Learning Center, Department of Medical Education, College of Medicine, University of Tennessee Health Science Center, Memphis, TN 38163, USA; ahall32@uthsc.edu; 3Department of Interprofessional Education, College of Graduate Health Sciences, University of Tennessee Health Science Center, Memphis, TN 38163, USA; dscott50@uthsc.edu; 4Teaching and Learning Center, University of Tennessee Health Science Center, Memphis, TN 38163, USA

**Keywords:** public health, pharmacy education, student pharmacist, endocrine disrupting chemicals, role playing

## Abstract

Preparing the next generation of pharmacists to succeed in practice and provide premier care starts with ensuring pharmacy education standards are met and align with innovative practices and that education incorporates topics that are important to promoting health. For example, recent reports link endocrine disrupting chemicals (EDCs) to numerous diseases such as reproductive disorders, metabolic diseases, and developmental abnormalities. Considering the suboptimal awareness and knowledge about EDCs, it is imperative to provide public health education through a pharmacy curriculum. The objective of this study was to evaluate student pharmacists’ perceptions of the impact of a role play activity on their knowledge of EDCs and counseling skills. A secondary objective was to explore student pharmacists’ perceptions of how role play might impact their future career as a pharmacist. A retrospective qualitative study consisting of a lecture, a pre-brief, a low-fidelity simulation centered on role-play, and debrief to develop knowledge of EDCs to practice counseling skills, and a post reflection was implemented to explore this aim. Third year student pharmacists who were enrolled on the public health elective course were eligible to participate in the study. All reflections were de-identified, imported into a qualitative software, Dedoose^®^, Version 9.2.6 and thematically analyzed using an inductive approach. Thematic analysis revealed three master themes, which tell the story of an initial lack of familiarity with EDCs that was rectified by the lecture and low-fidelity simulation. In the first theme, we can see that all of the participants noted their positive perceptions of the low-fidelity simulation, especially the role playing on a topic they lacked familiarity with. In the second theme, participants revealed the activity’s impact on their performance or behaviors. Finally, the third theme explores the future implications of a pharmacist’s impact on public health. This novel study contributes to a growing body of literature on the impact of pharmacy education practices and instruction on public health. The findings suggest that pharmacy educators might consider incorporating role playing instruction for public health topics, EDCs, or topics not traditionally taught in the pharmacy curriculum.

## 1. Introduction

The role of a pharmacist has transitioned from traditional dispensing roles and shifted to become recognized as a medication expert and a valued member of the healthcare team with key roles in providing education and managing chronic diseases [1]. These roles are further extended by integrating pharmacy students into innovative real-world models thus establishing foundational skills to provide patient education and care, while having a significant positive impact on patient outcomes [1].

Preparing the next generation of pharmacists to succeed in practice by providing premier care starts with ensuring education standards are met and align with innovative practices and incorporate topics that are important to promoting health. The field of pharmacy has seen substantial changes over the past few centuries, evolving from the manufacturing and distribution of various medical products to including a diverse range of essential clinical services and public health duties. In the 21st century, pharmacists have a wide range of responsibilities [2]. These include improving the way drugs are delivered, customizing treatment for each patient, overseeing long term non communicable diseases, advising on the use of drugs and dietary supplements in relation to nutrition, and dealing with the possible negative effects of environmental pollution on human health [2]. The existing national and European Union legislation should include all the specified responsibilities of the contemporary pharmacist [2]. Recently, emphasis has been placed on the topic of social determinants of health (SDOH), and pharmacy education standards now include assessing and addressing the SDOH [3]. Integrating the SDOH into education aligns well with one of the Healthy People 2030 goals to “eliminate health disparities, achieve health equity, and attain health literacy to improve the health and well-being of all” [4]. Healthy People 2030 provides foresight and outlines specific objectives to promote health and prevent disease in the coming years. As one of the most accessible health care providers, pharmacists are perfectly positioned to play key roles in various public health initiatives [4]. Another overarching goal of Healthy People 2030 is the prevention of disease and injury [4].

Through patient education efforts, pharmacists and student pharmacists can highlight variances between actual and optimal health statuses and directly impact aims to improve the health of the nation [4]. However, for future providers to assess and address the gaps in healthcare, educators must ensure they adequately prepare students to do so [4,5]. A study examining both pharmacy and medical students’ knowledge and attitudes relating to public health showed that 98% of survey respondents agreed that public health initiatives are valuable to the American healthcare system [5]. The same study showed that both medical and pharmacy students recognized core public health topics in their curricula, and nearly 90% of responding students wanted more information on those topics [5]. 

In recent years, growing evidence has linked endocrine disrupting chemicals (EDCs) to various diseases such as reproductive disorders, metabolic diseases, and developmental abnormalities [6]. Exposure to EDCs can occur from consumption of contaminated food and water or even arise due to the use of personal or home care products [6]. A study analyzing the cost and disease burden of EDCs in the United States showed that the disease costs of EDCs were much higher in the USA (340 USD billion) compared to Europe (217 USD billion) [7]. In conjunction, the population health risks imposed by EDCs and the related disease burden with associated costs will create a public health issue. 

Concerns about EDCs are not exclusively a public health issue. Social determinants of health need to be addressed within health professionals’ education. One study investigated the link between racial/ethnic differences in the use of personal care products with associations to EDCs [8]. As pharmacists, it is important to be aware of possible health discrepancies in over-the-counter products, so that recommendations for such products can enhance patient safety and overall health outcomes. There is a considerable need for public education and proactive prevention from exposure to EDCs [7]. However, there is a gap in the literature regarding healthcare professional education concerning EDCs. The means of exposure may unknowingly be facilitated by those in healthcare, thus worsening disease states and counteracting the effectiveness of treatments [6].

Several healthcare providers may lack the knowledge of knowing what EDCs are and thus ignorant of the associated risks/consequences. Furthermore, risk–benefit calculations utilized in decision-making for patient medical needs may skew calculations. Although healthcare providers may lack knowledge on EDCs, this does not exonerate them from the ethical issues that involve failure to disclose the potentially harmful properties of EDCs in medical interventions, thus infringing on patient autonomy, informed consent, and the oath to do no harm [6]. To reduce the knowledge gap between pharmacists and public health issues, the Toxicology Unit at the University of Alcalá in Spain conducted a study via an innovative short-term curriculum on environmental toxicology. The goal of this curriculum was to familiarize future pharmacists with the focus on public health and environmental toxicology via theoretical and practical sessions. Furthermore, the objectives were for student pharmacists to be able to recognize risks associated with environmental contaminants and to utilize applicable resources to employ basic actions to purify and recondition polluted areas affected by chemical compounds and/or drugs. In this study, Peña-Fernández et al. concluded that additional understanding and awareness of toxicology is essential for pharmacy and pharmaceutical industry professionals. This can be achieved through pharmacy-specific specialized training sessions (theoretical and practical) on environmental toxicology specifically [9]. While the study by Milanovic et al., 2023, touches on the effects of lower concentrations of EDCs in animal models, the authors conclude that an assessment of the rate of developing diabetes following exposure at lower levels is unknown [10]. Addressing education deficits in healthcare providers can lead to a clear identification of medical components/devices and medication products manufactured from EDCs. Additionally, clear labeling can assist with patient understanding and counseling on EDCs in order to assess risks and make confident knowledge-based medical decisions about their treatment options [6]. Haruty et al. constructed a series of tips that providers can utilize in counseling patients on reducing their risk of EDCs exposure. Tips included: (1) to reduce bisphenol A (BPA) exposure for infants, (2) to minimize pesticide contact at home, and (3) to avoid phthalates contained in many personal-care products [11]. To provide proficient care, specialized education on and recognition of the EDCs risks that can be accredited to everyday living encounters and those imposed by the healthcare professionals is warranted and ethically vital [6]. The objective of this study was to evaluate student pharmacists’ perceptions of the impact of a role play activity on their knowledge of EDCs and their counseling skills. A secondary objective was to explore student pharmacists’ perceptions of how role play might impact their future career as a pharmacist. 

## 2. Methods

### 2.1. Study Design

A retrospective qualitative study was designed to evaluate student pharmacists’ perceptions of the impact of a role play activity on their knowledge of EDCs and their counseling skills. A secondary objective was to explore student pharmacists’ perceptions of how the role play might impact their future career as a pharmacist. A thematic analysis was applied to identify common themes amongst student responses. 

### 2.2. Setting and Participants

This study was conducted at the University of Tennessee Health Science Center in the Doctor of Pharmacy (Pharm.D.) program, which comprises a four-year curriculum, including both didactic courses and experiential education. Based on the Accreditation Council for Pharmacy Education (ACPE), the vision of the Council of Credentialing in Pharmacy (CCP) is that “all credentialing programs in pharmacy will meet established standards of quality and contribute to improvement in patient care and the overall public health” [12]. 

During the Pharm.D. curriculum, students can enroll in various elective courses that explore different practices of pharmacy. While some elective courses focus on specific disease states such as clinical toxicology, infectious diseases, pediatrics, and critical care, other elective courses available offer exposure to niche areas of pharmacy practice in rural areas and public health. Student pharmacists in their third year are eligible to take the Public Health elective, which is offered in the Fall of each academic year. Furthermore, student pharmacists registered for the Public Health elective course receive an overview of public health principles, global health strategies, and pharmacists’ roles in public health at the state, national, and global levels.

During the 2022 and 2023 courses, eleven pharmacy students enrolled on the Public Health elective course and were all working simultaneously as pharmacy interns at the time of this course. This activity was conducted as part of required coursework within the elective and identified retrospectively for research. As such, informed consent was not required by the University of Tennessee Health Science Center Institutional Review Board. Purposeful convenience sampling was employed to obtain information-rich data using all student reflection responses [13]. No students were excluded. The Institutional Review Board approved this study as exempt (IRB# 23-09700-XM).

### 2.3. Procedures

During one two-hour class session held via Zoom, students participated in a combination of activities including a lecture, pre-brief, low-fidelity simulation centered on role-play, and debrief to develop knowledge of endocrine disruptor chemicals (EDCs) and to practice counseling skills. At the beginning of the session, a 50 min lecture was delivered to students that focused on (1) understanding the effects of EDCs on the neuroendocrine system of the human body and (2) addressing mechanisms behind EDCs as a public health concern. At the end of the lecture, the course director facilitated a 10 min pre-brief that introduced students to the low-fidelity simulation and case scenarios. After completing the lecture and pre-brief, student pharmacists were randomly assigned into break-out rooms to participate in one-on-one role play simulations focused on counseling patients and families about EDCs. This activity lasted for 25 min. Each breakout room included two students, except for one which had three students due to an uneven class roster. In addition, a faculty member joined each room to facilitate the activity and provide redirection when necessary. 

Groups were given a role play simulation guide during pre-brief, which included instructions, two case-based scenarios that consisted of two roles each, a pharmacist and a pediatric patient’s parent, a description of each role, and helpful hints on counseling and EDCs. After completing the role play of the first scenario, students switched roles for the second scenario. The group with three students repeated one of the scenarios allowing all three students the opportunity to play both roles. Keeping in mind that EDCs are complex, heterogeneous groups of chemicals, it is very difficult to provide general council about sources, possible effects, and how to limit exposures. During the role play simulation the pharmacist was charged with providing counseling to the patient’s parent regarding EDCs, including their source, possible effects, and how to limit exposure. Each student was instructed to use communication strategies either in the role of a future pharmacist or patient in both scenarios. After the role play activity concluded, the entire class reconvened to participate in a 20 min, faculty-led debrief using the Socratic method of open-ended questions that facilitated extemporaneous thought and conversation to flow. Then, students were assigned an out-of-class, three question, written reflection about their experiences during the role play activity, how it might influence their thoughts about EDCs, and how it might impact their future practice as a pharmacist. 

### 2.4. Data Collection and Analysis

The written reflection activity, completed as a word document, was submitted via email to the course director. All reflections were de-identified to provide anonymity to responses. Themes and subthemes were identified using thematic analysis with an inductive approach [14]. Data analysis followed the reflexive process to conduct a thorough and transparent data analysis. After familiarization with the data, the reflections were coded with verbatim pieces of text and then preliminary codes were generated. Similar codes were grouped into categories [14]. All categories were clustered and analyzed to uncover the major themes. Two researchers performed a line-by-line reading of the reflections, after familiarizing themselves with the corpus of data (AH, AC). After review, the researchers collaborated to define and refine the initial codes as well as identify emergent codes [14].

A process of independent review and open discussion was implemented to clarify and revise the coding frame until consensus was reached [14]. The coding process was facilitated by a qualitative software, Dedoose^®^ (v2.0, Manhattan Beach, CA, USA), which was used to generate initial codes and develop and review themes. Another researcher (DS) reviewed the final codes and one researcher identified common and recurrent themes and subthemes.

## 3. Results

The thematic analysis revealed three master themes, which tell the story of an initial lack of familiarity with EDCs that was rectified by the lecture and low-fidelity simulation. 

In the first theme, we can see that all of the participants noted their positive perceptions of the low-fidelity simulation, especially role playing on a topic they lacked familiarity with. In the second theme, participants revealed the activity’s impact on their performance or behaviors. Finally, the third theme explores the future implications of a pharmacist’s impact on public health.

### 3.1. Theme 1: Positive Reception of the Lecture and Low-Fidelity Simulation

The entire cohort of participants talked positively about the EDCs lecture. Additionally, participants highly recommended continuing to expose future classes to this topic and activity as part of the course. For example, the following excerpts highlight these findings in their individual reflections:

“I definitely think you both should continue this activity in this class as it was both fun and informational at the same time.” (2023)

“I enjoyed the lecture we had about Endocrine Disrupters. I found this lesson to be very informative.” (2022)

“The role-playing activity introduced towards the end of the class was insightful and productive.” (2023)

Another participant noted that the lecture and low-fidelity simulation helped them to recognize their limited knowledge about the topic. The participant summarized this information in the excerpt below. 

“I believe that by doing activities like this, it allows for me to recognize the areas of information I may not know and then seek out more information.” (2023)

#### Sub-Theme: Knowledge Gap

In their reflections, the student pharmacists also noted that they did not have any previous knowledge about endocrine disrupting chemicals. Some participants attributed their lack of EDCs knowledge to the absence of didactic lectures on the topic. The following excerpts summarize these findings:

“While role playing, I found it interesting how little I previously knew about the endocrine disrupters.” (2023)

“After the lecture, I completed some research on my own about endocrine disruptors. I will explain to [patients] the products to avoid like pesticides and plastics.” (2022)

One participant used the word “overwhelmed” when describing their feelings on how to handle the counseling points. Struggling with counseling could be taken positively because it is a learning opportunity. The student pharmacist powerfully summarized this struggle:

“Since the topic is new and the scenario was given ten minutes before the role play activity, I was completely overwhelmed to counsel the role-playing parent. It is interesting to see the struggle of role-playing pharmacist and parent to absorb the topic of endocrine disruptors.” (2022)

Another participant also expressed the feelings of struggling with counseling on a novel topic, a sentiment echoed in the following quotation: 

“I was surprised to discover that, when playing the pharmacist, I was not as at ease as I should be while counseling a patient as a [third year Pharmacist student]. I believe that this practice helped me to become more attentive and improve my counseling techniques.” (2022)

### 3.2. Theme 2: Impact of the Low-Fidelity Simulation on Participants

In this second theme, many participants noted that the low-fidelity simulation had different impacts, including increasing their investment in the role, reinforcing the counseling techniques they used, and leading to positive changes in their daily behavior.

For one participant, the low-fidelity simulation reminded them to be mindful of using complex medical terminology when counseling patients. 

“When I was portraying the mother, I was astounded at how someone could be so concerned and unaware of medical terminology. This action clarified the role those endocrine disruptors have in our general health. I am eager to learn more about this topic so that I can be prepared to counsel on it confidently when I am in clinical practice.” (2022)

Another participant highlighted the fact that the EDCs are ubiquitous in quotidian life. The participant stated:

“These scenarios impacted my thinking about endocrine disruptors by showing me how prevalent they are in our everyday lives.” (2023)

#### 3.2.1. Sub-Theme: Investment in Role

Several participants pointed out that the low-fidelity simulation had implications for applying the knowledge to their role and their personal understanding of the information. For example, one student pharmacist asserted the applicability of the intervention to daily experiences.

“The scenarios helped me apply the knowledge that we learned in lecture into a real-life scenario. Applying the knowledge in a practical way helped me solidify what I learned earlier.” (2022)

Finally, the below excerpts wrapped up the participants’ reflections surrounding the importance of EDCs by emphasizing EDCs’ role in everyday activities.

“It impacted my thinking about endocrine disruptors quite a bit. I truly had no idea the role it in play in our lives. How it impacts us and how much plastic we use every day.” (2023)

“I was surprised with how invested I became in my “children” when playing the mother.” (2022)

#### 3.2.2. Sub-Theme: Emphasis on Counseling Techniques

Several participants reinforced the importance of using appropriate counseling techniques to convey a message to the patient. For example, in the following excerpt the participant used strong words such as “does not scare them” in the sense that using appropriate terminology is pivotal in counseling and ensuring patient safety.

“This activity also reminded me how important it is to properly counsel patients in a way that does not scare them more, but alleviate their concerns.” (2023)

One can see how this participant’s emphasis was on the EDCs and child development. The participant commented:

“In the pharmacist role, I was surprised by how much influence a documentary had on patient thinking. With the access we have to media in this age, the public can be so quick to consume or dismiss things that they hear. However, the patients I interacted with in this scenario were very concerned on the possible implications of endocrine disruptors on their children’s development. I was happy that I was able to ease some of their worries by providing practical changes that they could start today to limit any risk.” (2023)

#### 3.2.3. Sub-Theme: Change in Personal Behavior

Several participants highlighted how impactful the information provided would be on their behavior. For example, the following quotes vividly demonstrate the change in individual behavior exemplified by using fewer plastic products.

“This lecture and role playing has inspired me to take a look inward and make more conscious decision when I can about move away from plastic and other materials.” (2023)

“I will use what I learned about endocrine disruptors in my future practice as a pharmacist by going to a more natural and holistic approach when it comes to the things I use and consume. I will have to be intentional about practical things I can do daily to live a better life. It’s the small changes that make a difference. I made small changes of using glass for my meal preparation and drinking out of glass bottles instead of plastic bottles as much. I learned about this topic via social media, and it made me dive deeper into learning about endocrine disruptors. Eventually, I want to start making my own detergent without all the chemicals. I can also do a better job at recycling.” (2023)

### 3.3. Theme 3: Future Implications for Public Health

In this theme, participants explored the future implications of a pharmacist’s impact on public health. For example, one participant commented that their key takeaway from the class has implications for the public health arena. The student pharmacist asserted the importance of key stakeholders in addressing EDCs and decreasing their usage. The participant stated:

“I enjoyed the scenarios a lot. It allowed me to put certain things in perspective and think quickly on my feet. The scenarios impacted my thinking about endocrine disruptors by allowing me to think practically about how the United States can do a better job at regulating endocrine disruptors and lessen our exposure to them.” (2023)

The following excerpts assert that patients trust pharmacists, and pharmacists need to focus on building trusting relationships with their patients to serve their community better. 

“As the patient, I realized how much trust patients place in the pharmacists. They will trust their information is accurate and will believe it as truth. The scenarios gave me a new insight into the patient’s concerns about endocrine disruptors. I can now think about how to counsel patients on endocrine disruptors and giving them the most accurate information in a way that is easy to understand.” (2022)

“It may seem daunting at first, but with education and empathy, I believe that these interactions can help continue to build rapport within the community.” (2023)

In the quotes below, a student pharmacist echoed the link between pharmacist and patient that leads to building a trusting relationship. Ultimately, student pharmacists highlighted that pharmacists must overcome obstacles and build trusting relationships with patients to successfully create a proactive community that is aware of public health concerns.

“It’s better to be proactive and not wait until something happens. I also think it’s a topic that we as healthcare providers can do a better job at learning and informing others about whether it is our patients, family members etc.” (2022)

“These scenarios allowed me to become more aware of the concerns of the community, which prompted further research.”(2023)

## 4. Discussion

Students’ reflections after the EDCs activity centered on three major themes: (1) positive comments about the effectiveness of the lecture and low-fidelity simulation; (2) the ways in which the EDCs activity will inform student pharmacists’ behavior as pharmacists moving forward; and (3) the public health responsibilities of pharmacists in terms of endocrine disrupting chemicals. Taken together, the students’ reflections demonstrate the educational possibilities of the lecture, role play, and reflection hybrid instructional model to guide student pharmacists to think deeply about previously underdeveloped topics and their role as pharmacists in the context of public health and patient counseling. 

Traditionally, lectures are used most frequently to impart knowledge to students in the healthcare professional curriculum, including pharmacy education [15]. During the lecture component of the EDCs activity, the student pharmacists were informed about EDCs and their potential public health implications [16,17,18,19,20,21,22].

The sub-theme on the EDCs knowledge gap demonstrates that the lecture component of the EDCs activity successfully expanded the students’ knowledge on a previously unknown subject area. Furthermore, several students identified EDCs as a subject area they would devote time to learning more about, to enhance their capabilities as a pharmacist and public health advocate. Student pharmacists reflected positively on the EDCs lecture and spoke about the ways in which it informed their participation in the role playing activity. 

This activity exposed the student pharmacists to role play and reflection to engage them during class and facilitate their critical thinking, problem solving, and counseling skills. As demonstrated by student pharmacists’ reflections, the role playing aspect of the EDCs activity helped to cement knowledge about EDCs and led to critical thinking about the role of pharmacists in the context of EDCs and public health and patient counseling. This outcome tracks with medical education research that indicates that role playing “with direct observation and feedback, proved efficacious in changing and improving interview behaviour” [23]. Furthermore, a recent literature review on role playing in mental health education found evidence to suggest that “role-play enhances students’ therapeutic and communicative skills” [24]. 

In theme two, student pharmacists reflected on counseling patients in the context of the role play activity and indicated that the activity improved their understanding of their patients and informed their counseling techniques positively. These reflections are aligned with educational outcomes associated with role-play and simulated patients. In a review of the literature on simulated patients and role play, Lane and Rollnick found that students experienced “positive improvements in the use of communication skills after conducting role-plays or consulting with a simulated patient” [25]. Moreover, Bharti et.al., 2023, argued that “Role play simulates real life scenarios and can help students apply their skills and knowledge acquired through role plays into practice when faced with similar situation in the community and clinical set ups” [26]. In their reflections, the student pharmacists tied their thoughts on counseling patients directly to the role play activity, indicating that this component of the EDCs activity was inexorably intertwined with their thoughts on the way the activity informed their thinking on patient counseling moving forward.

To enhance student pharmacists’ knowledge, develop clinical judgement, and enhance counseling skills, activities such as role play could be perceived as an effective learning modality [27]. The effectiveness of the role play components of the EDCs activity in stimulating critical thinking and reflection on patient interactions tracks with health sciences’ education research on the expected outcomes from role playing activities. For example, Vizeshfar et al., 2019 argue that “role-play is an efficient method for transferring knowledge and experiences to various health sciences groups” [28]. Ultimately, through the three themes, the student pharmacists reported that the lecture and role playing components of the EDCs activity worked in tandem to deepen knowledge of EDCs while simultaneously leading to critical thinking about the pharmacist’s counseling role in relation to EDCs. 

The final, written, student pharmacists’ reflections on the EDCs activity were an integral part of the educational experience, as they were designed to foster engagement with the public health implications of endocrine disrupting chemicals. Boud et al. (2013) define reflection as “a generic term for those intellectual and affective abilities in which individuals engage to explore their experiences in order to lead to new understandings and appreciations” [29]. Previous studies demonstrated that self-reflection skills incorporated in healthcare education are perceived as an effective active learning strategy that assists students in developing clinical decision-making skills and capabilities, and ultimately augments students’ academic performance [30,31]. 

Scholars posit that reflection is a crucial element of effective learning, as it fosters deeper learning and encompasses a critical assessment of learners’ beliefs which could result in an enhanced ability to make decisions [32,33,34]. Furthermore, reflection is an instructional approach to connect the gap between learners’ thoughts and actions, as it enables them to determine whether they have a sufficient grasp of the concepts, principles, and skills to be able to transfer them to a different environment [35]. Hence, post activity reflection is considered useful in the context of teaching and learning about a new concept such as EDCs. The literature contains various examples of the positive effects of using reflective learning in pharmacy curriculum. For example, developing reflective writing skills could be linked to enhanced academic performances [36,37]. In the context of this study, the positive impact of the reflection portion of the EDCs activity is highlighted in theme three, wherein student pharmacists reflected on the public health implications of EDCs. Student pharmacists demonstrated a deep understanding of the potential harms of endocrine disruptors when they reflected on the role of pharmacists in informing the public about the health implications of EDCs and the role of the United States government in countering the ill effects of EDCs on public health. As demonstrated in the reflections, the student pharmacists highlighted the connections that they made between the lecture, role play, and reflection portions of the EDCs activity, showcasing the development of their knowledge of EDCs and their thoughts on the applicability of their learning. 

The three themes identified in students’ reflections inform educational approaches to endocrine disrupting chemicals, public health, and pharmacy curriculum topics writ large. The three themes speak to the positive reception of a targeted EDCs lecture, student appreciation for the role playing activity’s ability to concretize their thoughts on EDCs, and the written reflection’s ability to foster critical thought on the public health implications of EDCs. The themes suggest that, in the course of one two-hour class activity coupling traditional lecture, role play, debriefing, and reflection, considerable educational outcomes can be achieved. Student pharmacists can progress from extremely limited knowledge of EDCs to not only understanding EDCs but actively reflecting on the public health implications of EDCs and the role of pharmacists in educating patients about EDCs. Furthermore, the EDCs activity may serve as a blueprint for teaching student pharmacists about public health content or topics that are not usually covered in the traditional pharmacy curriculum. Student reflections on the EDCs activity suggest that carefully partnering lecture, role-play, debriefing, and reflection can lead to a positive educational experience that encourages critical thinking and deep learning. 

### Strengths and Limitations

A novel aspect of this study is that we aimed to understand how to better equip student pharmacists with communication skills to address a broader spectrum of patients. For this reason, the team selected role playing, a technique that is not often employed in the Pharm.D. curriculum. Thus, a strength of this study is its introduction of the role playing educational strategy to pharmacy educators that wish to enhance student pharmacists’ understanding of how to counsel on novel topics and provide strategies to develop scenarios that resemble complex patient cases.

Another strength of this study was that the methodology combined a didactic lecture, role-playing, debriefing, and in-depth written reflections. This approach guides students to make connections between the lecture material, the scenarios, and, through the reflection activity, their lived experiences. Furthermore, this coupling of lecture, role playing, and reflection may inspire educators to incorporate a similar approach in their courses.

A limitation of this study is that it was conducted at one institution, and the course was offered as an elective. Although this limitation hinders comparative analysis between institutions, the value of conducting a qualitative study was that it gave access to the student pharmacists’ opinions through in-depth reflections. Additionally, the role-play activity required considerable time commitments which makes it challenging to achieve meaningful outcomes. Standardized patients report varying levels of comfort with the role-play process leading to a high demand for training and educational materials that might limit the resources available to other students participating in role play activities. Due to the costs associated with hiring standardized patients and the training required, efforts to develop cost-effective alternatives for this process might be considered. Finally, another limitation of the study is the potential for biases in the responses because they were collected as part of the course requirements.

## 5. Conclusions

The endocrine disrupting chemicals activity and study contribute to a growing body of literature on pharmacy education practices and instruction on public health in the context of endocrine disruptors. Pharmacy education practices on endocrine disruptors and other topics that are not usually covered in the pharmacy curriculum may be informed by the three themes identified in this research: (1) Positive reception of the lecture and low-fidelity simulation; (2) the impact of the low-fidelity simulation on participants; (3) future implications for public health. The student pharmacists’ reflections support the application of a hybrid of lecture, role play, and reflection instructional modalities in tandem to enhance student understanding and critical thinking on a previously untaught topic. Furthermore, the scholarship of teaching and learning research supports the use of each of these educational approaches in public health education. The results of this study suggest that pharmacy educators should contemplate modeling instruction on public health topics, endocrine disrupting chemicals, or topics not traditionally taught in the pharmacy curriculum on the lecture/role play/reflection hybrid model illustrated in this research. In the context of this research, student pharmacist reflections indicate that the lecture/role-play/reflection hybrid model may lead to a deeper understanding of the content area, enhanced awareness of patient counseling behaviors, augmented critical thinking, and heightened reflection on the public health role of pharmacists.

## Data Availability

Data is not available due to privacy and ethical restrictions.

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
