# Peer review of "Empowering Student Pharmacists to Counsel Patients on Endocrine Disrupting Chemicals through Interactive Role-Play"

_pharmacy, 2024, doi:10.3390/pharmacy12020055_

Round 1

Reviewer 1 Report

Comments and Suggestions for Authors

1. I am a bit confused at the level of student that were enrolled in the study.  Does the fact that they were interns indicate the final year of study? It says 3rd year for the public health study but the intern portion is confusing. 

2. I am happy to see the IRB approved the study but odd that consent wasn't required. 

3. The objective was to characterise student pharmacists' perceptions (notice the apostrophe is in the correct spot) but the results don't match - some of the results were about the role playing activity and its acceptability and impact rather than EDCs.  Might need to expand the objective. It seems the paper has a very pharmacy education feel to it rather than just the perceptions of students about EDCs. It might need to be reframed in this light.

4. Were the numbers of students included sufficient? How do you know?  

Author Response

  1. I am a bit confused at the level of student that were enrolled in the study.  Does the fact that they were interns indicate the final year of study? It says 3rd year for the public health study but the intern portion is confusing. 

Response: Thank you for asking for clarification. The term “intern” indicates that a student is enrolled in a PharmD program regardless of the year they are in of pharmacy school in the state of Tennessee. An intern is working under the direct supervision of a registered pharmacist. Some responsibilities include counseling, inventory management, and transferring prescriptions.

  1. I am happy to see the IRB approved the study but odd that consent wasn't required. 

Response: Thank you for asking for clarification. We provided the IRB approval for this study that was exempt.

  1. The objective was to characterize student pharmacists' perceptions (notice the apostrophe is in the correct spot) but the results don't match - some of the results were about the role playing activity and its acceptability and impact rather than EDCs.  Might need to expand the objective. It seems the paper has a very pharmacy education feel to it rather than just the perceptions of students about EDCs. It might need to be reframed in this light.

Response: Thank you for the valuable feedback. We revised the manuscript according to the expanded objectives. “The objective of this study was to evaluate student pharmacists’ perceptions of the impact of a role play activity on their knowledge of EDCs and counseling skills. A secondary objective was to explore student pharmacists’ perceptions of how role -play might impact their future career as a pharmacist. “

  1. Were the numbers of students included sufficient? How do you know?  

Response: Thank you for this suggestion. Literature regarding sample size in qualitative research does not provide a minimum number of participants for the results to lack error while maintaining statistical power as quantitative research requires (Creswell, 2015; Staller, 2021). Research recommends that sample size be based on research methodology and focused on identifying participants who can provide the most information-rich data (Creswell, 2015; Staller, 2021). As such, we chose a purposeful, convenience sample that included all students (n=11) who took the elective course. Because the purpose of our research was to look at student perceptions of a specific learning activity in which they had participated, adding more students to the study would not have gleaned relevant information.

Creswell, J.W. (2015). Educational Research: Planning, Conducting, and Evaluating Quantitative and Qualitative Research. Pearson.

Staller, K.M. (2021). Big enough? Sampling in qualitative inquiry. Qualitative Social Work, 20(4). https://doi.org/10.1177/14733250211024516

Reviewer 2 Report

Comments and Suggestions for Authors

Article by Cernasev et al., presents a worthwhile public health endeavor. Exposure to endocrine disrupting chemicals from environment and daily commodities has become a serious public health concern. Regulatory agencies are working for the advancement of risk and exposure assessment of EDCs.

However, I have few concerns and questions for the authors:

1.    Surely, pharmacist have an important role in public health and help people living a healthy life. There is no mention in the manuscript about the specific role of pharmacists in reducing the exposure of EDCs.

2.    According to the manuscript, this class on EDCs was in introductory class for most of the students. The manuscript lacks the further plan with this elective course. There should be a plan with application-oriented lectures. Most of the exposure to EDCs occur via daily commodities, packaged foods and environmental pollutants. There can be multiple career options for the pharma and toxicology students to help public health sector in the context of EDC exposure. Therefore, students’ application and skill-based training. So, without a further training plan this course doesn’t sound to be useful and promising.

I suggest including a descriptive course plan and goals with the training on EDCs in this manuscript.

3.    Please check the manuscript for typo errors and grammar. Example the abbreviation of EDCs is written as EDCS.  Manuscript should be thoroughly checked for the typo errors and grammar. 

Comments on the Quality of English Language

Manuscript should be thoroughly checked for the typo errors and grammar. 

Author Response

Surely, pharmacists have an important role in public health and help people living a healthy life. There is no mention in the manuscript about the specific role of pharmacists in reducing the exposure of EDCs.

  1. According to the manuscript, this class on EDCs was in introductory class for most of the students. The manuscript lacks the further plan with this elective course. There should be a plan with application-oriented lectures. Most of the exposure to EDCs occur via daily commodities, packaged foods and environmental pollutants. There can be multiple career options for the pharma and toxicology students to help public health sector in the context of EDC exposure. Therefore, students’ application and skill-based training. So, without a further training plan this course doesn’t sound to be useful and promising.

Response: Thank you for sharing your thoughts about the utility of the public health elective course at UTHSC. As course director I am open to feedback about improving the experience students receive in this course. Student pharmacists can select the public health elective as a requirement to fulfill the didactic responsibilities of the curriculum and is advertised as a course that will offer students an opportunity to “gain an understanding of public health theoretical frameworks and models to design research studies, discuss pharmacist-led public interventions in emergency crisis, and review of the epidemiology of selected disease states that pharmacists in public health practices.” As a requirement, students must complete evaluations of the course for the curriculum planning and course director to review to make meaningful and impactful changes moving forward. The feedback I have received is positive and has been shown to increase the chances of student pharmacists obtaining residencies and other positions they might not have received without the foundational exposure to this field. Furthermore, based on the Accreditation Council for Pharmacy Education (ACPE), the vision of the Council of Credentialing in Pharmacy (CCP) is that “all credentialing programs in pharmacy will meet established standards of quality and contribute to improvement in patient care and the overall public health.”  

  1. Please check the manuscript for typo errors and grammar. Example the abbreviation of EDCs is written as EDCS.  Manuscript should be thoroughly checked for the typo errors and grammar. 

Response: Thank you for catching this inconsistency; it has been addressed.

Reviewer 3 Report

Comments and Suggestions for Authors

GENERAL COMMENTS

I believe that the manuscript and the given results are informative and could be useful in order to improve education of pharmacy students and prepare them for the public health role of pharmacists. However, there are some concerns that cannot be ignored. Overall, font setting errors should be corrected.

  1. Abstract needs to be improved in order to better highlight the obtained results. The sentences given in lines 38-43 should be rewritten.
  2. Pharmacists in 21st century has important role in the health protection caused by environmental pollution (https://doi.org/10.2298/MPNS1712365N). More recent data should be provided regarding the types of EDCs as well as the associated adverse effects related to chronic exposure.
  3. Please use pharmacy students instead student pharmacists.
  4. Please highlight low-dose effects and nonmonotonic dose–responses of EDCs (line 118). For more details see doi:10.4239/wjd.v14.i6.705
  5. Line 129: instead “characterize” use “evaluate”
  6. Line 146: It should be underlined the total number of students at third year in order to evaluate if the number of 11 students statistically enough to make valid conclusions.
  7. A role play simulation guide can be provided as supplement material.
  8. Having in mind that EDCs are huge heterogeneous group of chemicals it is very difficult to give a general council about sources, possible effects, and how to limit exposures (line 176).
  9. Please try to be more precise in the section 2.4. Data Collection and Analysis, i.e. what categories.
  10. Line 387, quotation marks should be checked
  11. The part related to limitations should be expand!!!
  12. Line 456: The complete sentence should be rewritten.
  13. The title, discussion part and conclusion are not in a proper association. If the lecture/role-play/reflection hybrid model is the focus of the paper and EDCs are used just as a sample to test this model the title as well as introduction part should be rewritten.
Comments on the Quality of English Language

OK

Author Response

I believe that the manuscript and the given results are informative and could be useful in order to improve education of pharmacy students and prepare them for the public health role of pharmacists. However, there are some concerns that cannot be ignored. Overall, font setting errors should be corrected.

  1. Abstract needs to be improved in order to better highlight the obtained results. The sentences given in lines 38-43 should be rewritten.

Response: Thank you for the recommendation, and it has been addressed.

  1. Pharmacists in 21st century has important role in the health protection caused by environmental pollution (https://doi.org/10.2298/MPNS1712365N). More recent data should be provided regarding the types of EDCs as well as the associated adverse effects related to chronic exposure.

Response: Thank you for the suggestion. The study by Milanovic M. et. al., 2023 discusses this topic and has been addressed in the manuscript with supporting information. The citation will be included in the references.

  1. Please use pharmacy students instead student pharmacists.

Response: Thank you for the recommendation; however, this is the appropriate way to address students enrolled in an accredited U.S. college of pharmacy.

  1. Please highlight low-dose effects and nonmonotonic dose–responses of EDCs (line 118). For more details see doi:10.4239/wjd.v14.i6.705

Response: Thank you for bringing this to our attention. After further review of the article, additional supporting information has been added.

  1. Line 129: instead “characterize” use “evaluate”

Response: Thanks for the suggestions and it has been changed.

  1. Line 146: It should be underlined the total number of students at third year in order to evaluate if the number of 11 students statistically enough to make valid conclusions.

Response: Thank you for your suggestion; however, this is a qualitative study, and the statistical relevance is not applicable to this study. Additionally, student pharmacists can choose their electives from a list of options based on their interests, and availability, which leads to varying numbers of students enrolling in the public health elective.

  1. A role play simulation guide can be provided as supplement material.

Response: I agree this could provide utility, but unfortunately, the rights belong to the university and could present an issue with copyright.

  1. Having in mind that EDCs are huge heterogeneous group of chemicals it is very difficult to give a general council about sources, possible effects, and how to limit exposures (line 176).

Response: Thank you for the suggestion, and it has been added to provide additional information.

  1. Please try to be more precise in the section 2.4. Data Collection and Analysis, i.e. what categories.
  2. Line 387, quotation marks should be checked

Response: Thank you for catching this error- it has been addressed.

  1. The part related to limitations should be expand!!!

Response: Thank you for suggesting additional information. Following a deeper dive into the article, we were able to include more relevant information for this section.

  1. Line 456: The complete sentence should be rewritten.

Response: Thanks for this suggestion. Please see the revised limitations section where additional edits have been made.

  1. The title, discussion part and conclusion are not in a proper association. If the lecture/role-play/reflection hybrid model is the focus of the paper and EDCs are used just as a sample to test this model the title as well as introduction part should be rewritten.

Response: I appreciate your feedback concerning this and have proposed a title below in response: “Empowering Student Pharmacists to Counsel on Endocrine Disrupting Chemicals (EDCs) through Interactive Role Play.”

Round 2

Reviewer 2 Report

Comments and Suggestions for Authors

Thank you for addressing the concerns.

Author Response

Thank you for your time to review our manuscript.

Reviewer 3 Report

Comments and Suggestions for Authors

GENERAL COMMENTS

Authors have made improvement in revised version and manuscript has potential for publication, but in the current revised version some limitations are still present.

 Specific comments

  1. Pharmacists in 21st century has important role in the health protection caused by environmental pollution (https://doi.org/10.2298/MPNS1712365N). In introduction this is not highlighted.
  2. Part given in lines 158-161 is completely copied from 152-155. Please be aware that this part has to be rewritten.
  3. Check the font in lines 222-224 and include the relevant reference.

Author Response

Authors have made improvements in the revised version and manuscript has potential for publication, but in the current revised version some limitations are still present.

 Specific comments

  1. Pharmacists in 21st century has important role in the health protection caused by environmental pollution (https://doi.org/10.2298/MPNS1712365N). In introduction this is not highlighted.

Response: Thank you for your suggestion. We reviewed the article listed above and included relevant information to support the manuscript. 

  1. Part given in lines 158-161 is completely copied from 152-155. Please be aware that this part has to be rewritten.

Response: Thank you for your recommendation. We reviewed the article and addressed the comments above.

  1. Check the font in lines 222-224 and include the relevant reference.

Response: Thank you for your suggestion. We reviewed the article and addressed the comments above.